# *Sui Generis* Geographical Indications Fostering Localized Sustainable Fashion: A Cross-Industry Assessment

Sara Cavagnero [1] and Simona Giordano [2,*]

1    Business and Law Faculty, Northumbria University, Newcastle upon Tyne NE1 8ST, UK;
     s.cavagnero@northumbria.ac.uk
2    Department of Economic Geography, Aldo Moro University, 70100 Bari, Italy
*    Correspondence: simona.giordano@uniba.it

**Abstract:** As interest in sustainable fashion and localism mounts, there is a compelling need to foster purchasers' trust in claims made by fashion businesses. Geographical indications (GIs) have proven successful not only in reducing consumers' search costs through reliable labels but also in safeguarding identity and heritage and delivering added value for agricultural products. Building on the EU Commission proposed Regulation to protect craft and industrial products that rely on the originality and authenticity of traditional practices from their regions and drawing on the "fiber follows food" adage, this paper puts forward policy recommendations related to the proposed expansion of GIs to the fashion industry. Through cross-sector and transdisciplinary explorative research, this article provides evidence on how the origin link could be framed to accommodate apparel and footwear items within the scope of protection of the EU *sui generis* GIs system despite their "non-terroir" character. Key drivers and barriers to harnessing GIs' potential and enhancing the sustainability of localized fashion production are further explored based on the theoretical insights and comparative practical experience extrapolated from qualitative interviews with GI-protected winemakers in Apulia. Ultimately, the paper increases the understanding of the economic, ecological, social, and governance implications, which need to be addressed to improve the sustainability impact of *sui generis* GI systems before expanding them to the apparel and footwear domain.

**Keywords:** fashion; wine; geographical indications; consumer information; localism

## 1. Introduction

The dichotomy between the global and local dimensions of fashion has triggered much academic debate in the last decades, in conjunction with the "global-turn" that characterized the industry since the dismantling of the Multi-Fiber Agreement [1]. The issue has returned to the limelight more recently, as sustainability considerations have increasingly represented a key criterion to orientate brands' sourcing location decisions [2]. The pandemic contributed to uplifting attention to nearness and quality, enhanced cultural connections with the territory [3], and, to a certain extent, triggered a centripetal as opposed to a centrifugal movement, which is at the heart of the concept of localism [4].

The latter is often associated with sustainable development, drawing from the assumption that human and environmental flourishing depends on the health of inhabited ecosystems [5]. Geographical proximity influences what and how much is produced and consumed because the costs of each extra unit are borne in the same territory where people are established [6]. Hence, thinking and acting at the territorial level can contribute to sustainable development by improving the capacity of local stakeholders to control their resources and adjust their behavior accordingly [7]. Furthermore, by bringing together tangible and intangible natural and human assets, the territory can serve as a platform for new initiatives and act as a catalyst for meaningful change [8]. This implies not only perpetuating what already exists but also shaping strategic approaches and related adjustments for achieving sustainable goals.

Against this background, this paper builds on localization studies, where localization is defined as the "process of anchoring (or of uprooting) of resources and capabilities vis-à-vis a given society and biophysical environment" [9].

Although this phenomenon has been addressed in fashion studies for almost a decade, the influence of local actors on sustainable development strategies remains limited. In Fletcher's words, it appears that "designers, brands, policy-makers, environmentalists, wearers of clothes, have interpreted the radical potential of localism within what they know: business-as-usual, but with a bit more regional manufacturing" [6].

This consideration seems suitable for the legal framework as well. Despite the evocative power of the "Made in" [10], frequently used as a synonym for sustainability, the country-of-origin regulation seems ill-adapted to convey information about the impact of localized apparel production for two main reasons. Firstly, the "Made in" can be qualified as the "economic passport" of a product, as it identifies goods wholly produced in one country or, when two or more countries are at stake, the one where the most significant transformation occurred [11]. Secondly, it fails to reveal certain unique features of the manufacturing system, such as the small territorial scope of the value chain, amounting to products that are the result of territorialization processes [12]. In this frame, difficulties in communicating the "genius loci", i.e., "the spirit of the place", bound to the interactions among the historical, social, and spatial dimensions [13], received little academic scrutiny to date except for artisanal fashion production [14].

The same is not true for the food sector, which has for a long time focused on the concepts of proximity, witnessed by the so-called "locavores", i.e., people who search for food produced and sold in the same territory where they live [15], as well as on the "sense of place" resulting from the process of building space by delimiting, occupying, transforming, and differentiating it and leading to a revitalization of the local and the meaning given to it [16]. These phenomena led to the development of sustainable strategies in rural areas also through the establishment of labelling schemes for local and organic products, both at national and EU levels [17]. In particular, geographical indications ("GIs") have proven successful not only in reducing consumers' search costs but also in safeguarding identity and heritage and delivering added value for agricultural products [18]. This was recently acknowledged by the European Commission, which envisaged the elaboration of sustainability criteria for bolstering quality-led, place-based systems that enhance threatened environmental and social resources [19]. The Commission also proposed a draft Regulation for protecting non-agricultural goods based on the *sui generis* GI system currently applied to wines, spirits, other agricultural products, and foodstuffs [20].

Building on the aforementioned policy proposals and drawing on the "fiber follows food" adage [21], this paper addresses sustainability implications arising from the expansion of GIs to the fashion industry.

It explores under which conditions *sui generis* GIs can be a viable solution for the future of the apparel and footwear sector, which is more and more linked to ideas, stories, and practices at the local level. Key drivers and barriers to harnessing GIs' potential and enhancing the sustainability of the local fashion industry are investigated based on the theoretical insights and comparative practical experience of winegrowers who have gained a reputation for their biological and high-quality wine, which is protected through a GI.

Bringing together two fields that normally operate in silos, i.e., food and fashion, this cross-sector and interdisciplinary research gathers data from the wine industry and combines them with the findings on product specifications related to geographically rooted fashion items. Eventually, a well-reasoned conceptual framework that could be applied to *sui generis* GIs for textile and apparel is elaborated to achieve two concurrent objectives: firstly, to link a specific product to a given territory and secondly, to boost the sustainability of locally produced items.

## 2. Theoretical Background

### 2.1. The Potential of GIs for Geographically-Rooted Fashion Items: Re-Framing the Origin Link through the Socio-Terroir Concept

In the European Union, the link between an agricultural product and the territory forms the basis of GIs. Differently from "Made in", GIs identify products with specific features and reputations essentially attributable to the natural and human resources of a specific area, as defined in Article 22(1) of the TRIPS Agreement [22].

GIs stem from the concept of "terroir": considered almost untranslatable from French, the term defines a portion of land recognized for its agricultural and, above all, wine-producing potential, and reflects the connection between origin and quality [23]. The protection initially granted to the wine sector was extended to the entire agri-food industry and is currently used both in Europe and beyond for a wide variety of goods [24].

Opposite to agricultural products, fashion items are "non-terroir", being characterized by a reputational link with the manufacturing place or by reference to localized technical know-how [25]. As of 2022, no harmonized mechanisms safeguarding non-terroir items are in place within the EU, where over half of the member states grant protection to non-agricultural products via *sui generis* GI systems (designed specifically for the protection of geographically rooted products), while the others rely on trademark-based protection systems (i.e., general IP mechanisms that are not designed to protect specifically geographically rooted products) [26].

Since 2011, the European Union has considered the introduction of an EU-wide system for the protection of GIs for non-agricultural items [27]. The European Commission has been called on to create a regulatory framework for the protection of geographically linked industrial and handicraft products since 2013 [28]. In autumn 2015, the Parliament endorsed an initiative report on the possible extension of EU GI protection to non-agricultural products and called on the Commission to make a legislative proposal to that effect [29]. After several scoping studies [30] and the EU accession to the Geneva Act of the Lisbon Agreement on Appellations of Origins and Geographical Indications [31], in the IP Action Plan adopted on 25 November 2020, the EU Commission asserted that it "stands ready" to consider the introduction of a system protecting non-agricultural items [20]. More recently, in April 2022, the proposal for a Regulation on geographical indication protection for craft and industrial products was published [32].

This study maintains that the GI discourse can emphasize the influences of local customs, long-standing production, and marketing traditions of geographically rooted fashion items by building on the "socio-terroir" concept. Described as the result of complex interactions among a set of natural and human factors [33], socio-terroir elements would enhance historical anchoring and collective memory and assign meaning to territories and related products. As the community dimension becomes a core part of the local fashion culture, it is possible to distinguish the "nationality" (i.e., being "born" in a place) from the "provenance" (i.e., coming from a place and belonging to it), where the latter becomes crucial to reshape the "origin link" of apparel and footwear.

In light of the above, the first research questions this paper addresses are how the origin link is currently framed against the different legal mechanisms safeguarding apparel and footwear in place in the EU member states. This information will, subsequently, be relied upon to develop recommendations on how such origin link can be more coherently shaped, based on the socio-terroir concept, to suit sustainable fashion items.

### 2.2. The Link between GIs and Sustainability: Cross-Industry and Transdisciplinary Research between the Wine and Fashion Sectors

GIs have played a key role in EU agricultural policy, strategies for rural development, and, more recently, in sustainability action plans [34]. Indeed, besides their economic relevance, GIs contribute to preserving the local culture and traditions, are a cause of pride for those who engage in the production of certain goods, as well as a tool for consumers to identify and appreciate traditional local specialties [35].

The fact that GI schemes could contribute to sustainable pathways has been widely expressed by scholars [36] and recognized by the EU Commission in the Proposal for a Regulation on EU GIs released on 31 March 2022 [37].

However, recent literature stressed that sustainability issues are not adequately addressed in GI legislation at all levels [38]. Most amendments related to GI product specifications are driven by market factors, and when revisions occur for environmentally-related reasons, they mostly result in the introduction of more flexible rather than stricter rules [39].

Adhering to Belletti et al.'s reasoning [18], this paper argues that GIs can play a prominent role in activating and supporting sustainable development, especially at the local level. However, this effect is not automatic and requires appropriate regulation related to manufacturing and consumption. Considering the envisaged expansion in the scope of *sui generis* GIs to non-agricultural items as well as the proposal to introduce specific sustainability criteria related to GIs in a broader sense [19], we argue that the EU Commission now has the unique opportunity to conceive a comprehensive conceptual framework to link locally produced items with sustainability considerations.

To this end, the wine sector, where *sui generis* GIs have proven successful not only in reducing consumers' search costs but also in safeguarding identity and heritage and delivering added value, can be treated as a testing ground for evaluating concepts and ideas before transferring them to the apparel and footwear domain. Although fashion is a unique context, and many differences exist with commodity product categories given, among the others, the role that clothes play in people's lives and the complexity of sustainability claims being made [40], a cross-industry and transdisciplinary approach could stimulate reflexive learning about the relevant plurality of underlying values, perspectives, assumptions, and institutional and power structures [41].

Accordingly, the second question this research aims to answer is which social and environmental aspects already fall within the GI protection system and which ones could be integrated into product specifications to better address sustainability concerns. Furthermore, power relations, market organization, cultures, and traditions will be also investigated. The overall objective is not only to improve the understanding of the sustainability potential of *sui generis* GI but also to extrapolate effective solutions transposable to the fashion sector to inform policymakers.

## 3. Research Methodology

To address the research questions, the work is divided into two distinct parts following different methodologies.

First, exploratory desk research was carried out to identify how the origin link is currently framed in product specifications in the absence of a unitary EU framework and considering the evolution of the "terroir" concept. Secondary data were collected by relying on a pre-existing IP database, freely accessible online.

Subsequently, qualitative research was conducted to gain primary information on which social and environmental considerations characterize GI products per se and which ones should instead be integrated into product specifications to boost the sustainability credentials of protected goods. Interviewees are winegrowers who gained a reputation for their biological and high-quality wine and successfully protected and promoted it through *sui generis* GIs.

### 3.1. IP Assets Protecting Locally Made Garments and Footwear in France and Italy

The purpose of this section is to collect and synthesize secondary data on how the origin link is currently outlined against the variety of legal systems in place within the EU for the protection of geographically rooted fashion products.

As a first step and to achieve this objective, a choice of countries was conducted to form the basis for a comparative assessment. France and Italy were selected based on two main factors. First, they are the leading EU countries in the field of *sui generis* GIs for agricultural products [42]. Further, with reference to non-agricultural items, they depict the

complex variety of solutions adopted by the EU Member States. Indeed, in France, the legal protection of industrial and craft products via sui generis GIs was specifically introduced by Law No. 2014–344 [43], which added a new section to the IP code. Conversely, Italy relies on trademark-based protection systems, as it lacks an ad hoc origin scheme.

With reference to the French case, online desk research on the national GI database, administered by the National Institute for Intellectual Property (INPI) [44], was conducted in April 2021. Through the exploration, 11 registered GIs related to non-agricultural products were retrieved [45]. Among them, two referred to apparel and footwear, namely "Linge Basque" [46] and "Charentaise de Charente-Périgord" [47], and were therefore selected for this research.

Moving to the Italian framework, the analysis started from the assumption that, at the EU level, two trademark-based protection systems are theoretically available for national applicants for protecting non-agricultural items, namely the EU collective marks and the EU certification marks [48]. Both certification and collective marks are general protection systems that are, in principle, not designed to protect geographically rooted items but primarily serve the purpose of indicating the collective commercial origin of an item or of guaranteeing certain features of a product, respectively [49].

According to Article 74 EUTMR, EU collective marks define common characteristics that distinguish the goods or services of the members of the association that owns the mark from those of other undertakings. Only associations of manufacturers, producers, suppliers of services or traders, or legal persons governed by public law may apply for collective marks. With reference to the causal link with a specific territory, Article 75(2) EUTMR sets that the collective mark may refer to the geographical origin of goods or services provided that the regulations of use authorize any person whose goods or services originate in the concerned territory to become a member of the association that is the proprietor of the mark. Article 83(1) EUTMR defines EU certification marks as marks that are "capable of distinguishing goods or services which are certified by the proprietor of the mark in respect of material, mode of manufacture of goods or performance of services, quality, accuracy or other characteristics, with the exception of geographical origin, from goods and services which are not so certified". In other words, they showcase that goods or services, manufacturers, or providers have met certain standards set out in the regulations of use. Accordingly, they shall be adopted by undertakings other than the owner, whose permission to affix the mark is subject to fees and routine inspections carried out under the responsibility of the owner itself.

For the sake of this research, certification marks were excluded because the definition provided by Article 83(1) EUTMR explicitly prevents their use to certify geographical origin. Collective marks were instead studied given that they can designate the geographical origin of the goods or services that bear them.

Accordingly, research on the EUIPO eSearch plus database [50] was carried out in April 2021, with the following entries:

- Kind of mark: collective;
- Nice Class: 25 (including apparel and footwear);
- Trademark status: registered.

The inquiry led to 137 search results, which were further reduced to 22 based on the nationality of the applicant. Among them, 12 trademarks were excluded because they were registered by either touristic associations or local institutions for the sake of the promotion of specific areas in Italy. Two trademarks were further eliminated, as the applicant was a single company rather than a collective entity. Three more marks were excluded because they were meant to promote a particular type of feather, eco-leather, and thread, respectively, without any specific link to a territory.

The four remaining marks were then fully assessed. Since comprehensive information was not easily retrievable on the EUIPO database, integrative research via official websites of the examined schemes was conducted as well. All the considered marks identified

geographical origin together with the environmental and social conditions that characterize the manufacturing process.

Two marks were registered by "Genuine Italian Vegetable-Tanned Leather Consortium", with a different graphic representation, one of which is not used anymore (despite being registered). Therefore, only the one currently in use, namely "Pelle Conciata al Vegetale in Toscana" [51], was selected for the sake of this research.

The other two marks were registered by the "Prato-Pistoia Chamber of Commerce" and refer to carded wool. The "Cardato Recycled Made in Prato" mark [52] was preferred over "Cardato" for this research because more producers (six compared to two) adhered to the scheme, with it being linked to regenerated wool, which provided for over a century an opportunity for the development and growth of the textile district of Prato.

To guide the information research, for each case study, the following seven clusters of information were retrieved within the product specifications and subsequently examined:

- Data sourced and adapted from;
- Type of product;
- Production method;
- Raw materials;
- Nature of the link;
- Applicant;
- Check and balance mechanism.

Finally, to better assess and compare data, the information was organized into two tables, one for each country, depicting the findings of the two national case studies. Each table counts seven rows, embedding the information for all the clusters outlined above.

### 3.2. Sustainability and Sui-Generis GI: The Wine Industry in Gioia del Colle

Exploratory qualitative analysis was carried out to identify potential benefits deriving from GIs as far as sustainability is concerned.

The study involved 14 out of the 15 wine producers operating in Apulia, Southern Italy, who were granted the Gioia del Colle PDO [53]. Although one company decided not to join the research, the participation rate was high thanks to the positive intervention of the President of the PDO Consortium, who actively promoted this study during the monthly meetings with wine producers.

The territory and the GI were chosen for three main reasons: Firstly, in the last decade, the Gioia del Colle PDO repositioned in the national and international markets due to the renovated attitude and coherent actions aimed at highlighting the uniqueness of local production in terms of soil, climate, altitude, and grape variety [54]. Further, sustainable wine production through the traditional "Apulian alberello" system and slow oeno-tourism projects have risen, allowing the integration of primary (agriculture), secondary (wine industry), and tertiary (tourism) sectors, playing an important role in rural development. The success case in the wine sector also spawned the food industry, with Gioia del Colle being the only area in Apulia being granted two PDOs both for wine and for the locally produced "mozzarella" [55].

Among the respondents, four producers are structured in micro-, eight in small-, and two in medium-sized enterprises (pursuant to the Italian legal definition [56]).

During structured interviews, the respondents were requested to answer a questionnaire (not reported in this article to accommodate the word limit but available upon request) drafted in Italian, which includes 45 items (mandatory: 37 standardized, closed-ended items (dichotomous (yes/no) based on SDGs identified in Table 1; mandatory: 4 three-point Likert scale; optional: 4 open-ended (exploratory) questions) aimed at assessing benefits linked to the GIs' adoption. The exploratory qualitative items investigated organizational functioning and allowed establishing more intense interaction with the informants, drawing on multiple sources of information and collecting robust data [57].

**Table 1.** GIs for fashion items in France.

| | Linge Basque | Charentaise de Charente-Périgord |
|---|---|---|
| **Sourced and adapted from** | INPI-2003—**Linge Basque** https://base-indications-geographiques.inpi.fr/fr/document/linge-basque-0#ig-detail (accessed on 15 March 2022) | INPI-1901—**Charentaise de Charente-Périgord** https://base-indications-geographiques.inpi.fr/fr/document/charentaise-de-charente-p%C3%A9rigord#ig-detail (accessed on 15 March 2022) |
| **Type of product** | – Fabric; <br> – Finished product; <br> – Transformation of linen into finished product. | Finished product: closed shoe with an upper part that runs up the instep. Flat sole. No right or left foot. |
| **Production method** | Four mandatory steps: warping, knotting, setting, and weaving (on the loom). <br> Making up, finishing, and shipping operations are not mandatory for the fabric production. <br> Only traditional motifs and geometric patterns admitted. <br> No color limitations. | Sew-and-turn only, in four steps: <br> – Cutting of the components used for the upper part; <br> – Stitching the elements of the upper part; <br> – Assembling of the shoe; <br> – Finishing. |
| **Raw materials** | Natural fibres only: <br> – Half-linen: cotton warp/linen weft; <br> – Cotton: cotton warp/cotton weft; <br> – 100% linen, with specific wight and yarns. | No limitations for the upper and lining materials. <br> For the sole: <br> – Felt of French origin (composed of wool, cotton, and where appropriate and to a minor extent, other materials); <br> – Leather of EU origin—basane (lamb leather) and croute de cuir (full-grain leather split); <br> – An anti-slip coating may be applied as an outer layer. |
| **Nature of the link** | Production located in the area for centuries. <br> Traditional, cultural, and symbolic meaning of weaving. <br> Distinctive features of the linen (strength, durability) associated with the identity of the Basque people. <br> Soil and climate characteristics ensuring flax cultivation. <br> Strategic geographical position for supplying complementary raw materials. <br> Supply chain fully structured within the territory, including cotton spinning and industrial dyeing, because of the distance from the centers of production or processing of raw materials. <br> Shaping/adaptation to the needs of local populations from self-consumption to industrialization in connection with agro-pastoralism and agriculture. | Product stemming from vegetable fiber or leather deriving from local agriculture and livestock: <br> – Historical and geographical origin: the development of the paper industry brought together small weavers, wool dyers, and felt makers who created the first felt factories for paper mills along the rivers; <br> – "Cousu-retourné", "au point croisé", or "point de chausson" know-how maintained over time in the area, shaping social interactions; <br> – Supply chain established in the area; <br> – Reputation: comfortable and warm (promoted by Presidents François Mitterrand and Jacques Chirac). |
| **Applicant** | Syndicat des tisseurs du linge basque, established in 1953. | Association pour la promotion de la Charentaise, established in 1901. |
| **Check and balance mechanism** | Internal and external check-audit. | Internal and external check-audit. |

Primary datasets were gathered via email (10 respondents) or online video calls (4 respondents), depending on the availability of the companies, within December 2020 and January 2021. All answers were treated anonymously.

Data were sorted via a cognitive analysis by drafting a table and allocating the outcomes and the quotes of the wine companies' managers into a taxonomy within four core clusters:

1. Economic;
2. Environmental;
3. Social;
4. Holistic.

Finally, the findings were associated with each of the 17 goals set by the Agenda 2030 [58] and compared with the categories established in the evaluation support study on geographical indications, published by the EU Commission in 2021 [59]. New outcomes were derived by creating further sub-categories and then analyzed to gain further insights into how different rationales for the adoption of sustainable practices surrounding GIs have developed.

Eventually, two main columns portray, on the one hand, the "Straight-forward benefits", namely the sustainability characteristics that are within the *sui generis* GI system. On the other hand, the column "Ancillary advantages and required actions" illustrates the measures that can be taken to valorize social, environmental, or economic credentials through product specifications by laying down the related requirements.

## 4. Results

### 4.1. Terroir Criteria Safeguarding Localized Fashion Items

The exploratory research aimed at understanding how the origin link is currently framed for apparel and footwear products against each distinctive regulatory framework and considering the evolution of the terroir concept.

The results, summarized in Tables 1 and 2, reveal that despite the legal asset relied upon, in all cases, the product specification reflects the prominent role of socio-terroir elements as well as the interplay of place-specific knowledge and culture, traditional practices, and use of nearby resources, climate, and geographical position.

With specific reference to the examined clusters, the following information emerged.

### 4.1.1. Type of Product

Both *sui generis* GIs in France and collective marks in Italy refer to finished products, fabric, yarns, and manufacturing processes.

### 4.1.2. Production Methods

The relevance of each manufacturing step and its location varies, encompassing mandatory requirements, more general guidelines, and the depiction of customary habits. Overall, the French GIs provide much more detailed descriptions compared to the Italian marks. This does not seem to be connected with the legal requirements, as the French law does not provide for mandatory rules on the steps that must occur in the identified territory. The only requirement is that producers comply with the product specifications approved by the National Institute of Intellectual Property, establishing which production and transformation processes must take place in the designated area to ensure the characteristics of the good.

### 4.1.3. Raw Materials

While the type of fiber is always thoroughly described, the link with the sourcing location is generally loose, with the only exception of Pelle Conciata al Vegetale.

**Table 2.** Collective marks for fashion items in Italy.

| | Cardato Recycled Made in Prato | Consorzio Vera Pelle Italiana Conciata al Vegetale |
|---|---|---|
| **Sourced and adapted from** | **Cardato Recycled Made in Prato** http://www.cardato.it/it/marchi/marchio-cardato-recycled/ (accessed on 15 March 2022). | **Consorzio Vera Pelle Italiana Conciata al Vegetale** http://www.pellealvegetale.it/en/consortium/ (accessed on 15 March 2022). |
| **Type of product** | "Lana meccanica": regenerated wool fiber resulting from the carbonization and shredding processes of used or discarded fabrics, rags, or scraps of clothing. Fabrics and yarns deriving from the transformation of mechanical wool, being composed at least of 65% "lana meccanica". | Leather products, as defined by Directive 94/11/CE, transformed through vegetable tannins. |
| **Production method** | Not regulated. | Not mandatory but traditional and generally abided by:<br>– Processed in pieces, with varying thicknesses (0.7/0.8–4.0 mm) depending on the final product;<br>– Tannins extracted from chestnut, mimosa, and quebracho trees. |
| **Raw materials** | At least 65% of used or discarded fabrics, rags, or scraps of clothing. | Leather originates from Tuscany; if not, it has undergone the entire manufacturing process of most relevant phases in Tuscany. |
| **Traceability** | The company is required to trace the production history of the goods, with particular emphasis on:<br>– The type of materials and components;<br>– The history of its manufacture and distribution. | A leather list, including sourcing and manufacturing locations, shall be always available for auditing purposes. |
| **Sustainable impact** | Preliminary environmental impact assessment required throughout the product life cycle, from the acquisition of raw materials or the generation of natural resources to final disposal (UNI EN ISO 14044). | – EMAS (Eco-Management and Audit Scheme) certification;<br>– ISO 26000 (social responsibility) applies;<br>– The raw hides are the discarded by-products of the food industry (meat for human consumption);<br>– Biodegradable product at the end of its lifecycle thanks to the preservation of chemical-biological characteristics;<br>– Many of the substances used during the tanning process are recovered, recycled, and reused in different fields, e.g., hair removed transformed into agricultural fertilizer, and sludge produced by the depuration plants is repurposed as construction material;<br>– No toxic substances, such as azo-dyes, nickel, PCP, or chrome VI, which are harmful not only to man but also to the environment;<br>– Absence of heavy metals. |
| **Territorial link** | Prato district, defined according to Art. 36 of law 317/1991 modified by art. 6 paragraph 8, law 110/1999. | Associated companies must have their production facilities in Tuscany. Suppliers shall operate in Tuscany, and a list of suppliers involved in the production process shall be kept for audit purposes. |
| **Applicant** | Prato Chamber of Commerce | Genuine Italian Vegetable-Tanned Leather Consortium. |
| **Check and balance mechanism** | Consultancy Company Certification body carries out the inspection for data validation | Internal and external check-audit. |

This also emerged from the legal requirements of the French GI legislation, where raw materials are not mentioned in the black letter of the law. The only reference could be inferred from the vocabulary "extraction" and "elaboration", retrieved in Articles L721-2 and L721-7 No. 5, respectively. Besides the ambiguous reference to sourcing activities, it is also worth noting that Art. 721-7 No. 5 does not require the product specification to mention where these steps are to occur.

### 4.1.4. Link with the Geographic Area

This element largely differs in the two considered countries: while the French product specifications analyze in depth the historical, cultural, socio-economic, environmental, and geographical connections, in the Italian collective marks, these circumstances are barely mentioned in the Regulations of Use and could be retrieved only on the owners' websites. This does not entail that such a link is lacking. In the Prato case, the work of the "cenciaiolo" dates to the XII century AD and relates to the lack of raw materials, while leather production flourished in Florence since the XIV century, taking advantage of the abundance of water and woods to obtain dyes and the surplus of labor in agriculture.

### 4.1.5. Traceability

Contrarily to the French GIs, both Italian marks strongly focus on traceability as well as on environmental or social impact. In the Cardato case, the fulfilment of these conditions is a preliminary and mandatory requirement to apply for the collective trademark. The criteria to be met are very comprehensive and encompass the full lifecycle. The same is true for the Pelle Conciata al Vegetale, where the respect for the natural ecosystem is presented as inherent to the vegetable-tanning process.

### 4.1.6. Applicants

In all the examined cases, applicants are well-established entities that have operated in the territory and in connection with producers for a long time. This is especially striking for the French GIs, where the associations have been active for at least 70 years. Again, this does not seem to be related to the requirements set by the law, which, in terms of representativeness, sets very general requirements. It establishes that the GI is applied for and administered by a private collective body on behalf of all the local producers that acquired its membership. This entity is called to preserve and enhance the value of the local traditions and know-how as well as of the derived products.

### 4.1.7. Check and Balance Mechanism

In all examples, an internal and external control and sanction system are in place except for Cardato, where audits are conducted by the certifiers directly.

Interestingly, for the French case studies, such a lack seems to contradict the black letter of the law, which sets that the applicant is also in charge of performing audits to monitor compliance with the product specifications.

### *4.2. Sustainable GI Production from the Winegrowers' Perspective*

The qualitative analysis contributed to revealing GIs' influence on sustainability commitments for the Gioia del Colle PDO wine. The benefits, pitfalls, and required actions, summarized in Table 3, could be used as a reference to draw sustainable and local production together in *sui generis* GIs' product specifications.

The research confirmed prior studies, which highlighted that GIs do have an inherent sustainability potential, but its fulfilment is deeply entrenched with voluntary actions, which shall go beyond the legal minimum. Such actions could be formalized into mandatory requirements, operating as a pre-condition for obtaining or maintaining the GI.

**Table 3.** Potential contribution of GIs with reference to SDGs, based on interviews with Gioia del Colle winemakers.

| Sustainable Development Goals | GI's Potential Contribution | |
|---|---|---|
| | **Straightforward Benefits** | **Ancillary Advantages and Required Actions** |
| | **ECONOMIC IMPACT** | |
| **7 AFFORDABLE AND CLEAN ENERGY** | Energy from renewable sources (only); Boost in energy savings through more efficient strategies. | Costs and infrastructures to improve energy efficiency. |
| **9 INDUSTRY, INNOVATION AND INFRASTRUCTURE** | Access to financial services and diversification of funding sources; Greater investment in R&D; New marketing channels; International market segments reached; Implementation of innovative techniques. | Weak infrastructures (physical and digital) that need to be reinforced;New technologies or environmentally friendly industrial processes needed to compete in global markets;Urgency to enhance use of digital tools;Lack of economic incentives based on environmental commitment or quality. |
| | **SOCIAL IMPACT** | |
| **4 QUALITY EDUCATION** | | Boost cooperation with education system for training skilled workers (especially for manual activities);Financial support for training or for obtaining advisory services;Foster skills development through vocational training and innovation "on the job";Facilitate inter-generational transmission of traditional techniques (also in innovative ways). |
| **5 GENDER EQUALITY** | | Higher involvement of women in leading positions;Poor implementation of gender-sensitive policies;Increase number of women in managerial roles;Limited promotion of gender equality and women empowerment at all levels;Equal wage is still not always achieved. |
| **8 DECENT WORK AND ECONOMIC GROWTH** | Higher diversity in the workforce (especially nationality and ethnicity). | Positive role in local employment (job creation and maintaining) in rural areas;Better income than non-GI enterprises not confirmed;Non-systematic impact on income;Differences among territories and non-homogeneous benefits;Youth employment still needs to take off;Code of conduct: recognized importance but still low adoption. |

**Table 3.** *Comt.*

| Sustainable Development Goals | GI's Potential Contribution | |
|---|---|---|
| | **Straightforward Benefits** | **Ancillary Advantages and Required Actions** |
| | **SOCIAL IMPACT** | |
| **10 REDUCED INEQUALITIES** | Increased social, economic, and political inclusion within local communities; Discrimination (gender, race, ethnicity, origin, religion, etc.) is more effectively managed; Preservation of living cultural heritage, especially traditional techniques. | Social protection policies do not go beyond legal requirements, Higher vulnerability for seasonal workers and migrants. |
| | **ENVIRONMENTAL IMPACT** | |
| **13 CLIMATE ACTION** | | Awareness-raising campaigns needed;Low data and scattered actions to reduce GHG emissions;Waste prevention, reduction, recycling, and reuse not widely implemented;Little knowledge of profitable circular economy strategies;Productivity affected by climate change and no support to producers;Need to implement adaptation, mitigation, early warning, and impact reduction measures in business strategies but institutional support needed. |
| **14 LIFE BELOW WATER** | More efficient water-management strategies also due to traditional production methods and localized value chain. | Reduced use of chemicals and chemical fertilizers, especially affecting water: still not related to GIs' adoption;Impact on the production intensity after chemical removal;Limited awareness of marine pollution related to land-based activities. |
| **15 LIFE ON LAND** | Ancillary benefits of promoting cultural heritage and traditional landscape (e.g., restoring biodiversity). | Minimized impact of the value chains on environment (water, soil, air), improved animal welfare, preservation of traditional landscapes: not dependent upon the GI;Fostered biodiversity using old varieties and techniques but lack of data collection to support the statements;Favor for more extensive methods (imposing maximum yields);Slow process with different level of commitment depending on public and private initiatives not strictly related to the GI;Voluntary-based initiatives, lack of encompassing and binding requirements;Conservation, restoration, sustainable use of terrestrial, inland freshwater ecosystems (forests, mountains, drylands): boosted following the independent adoption of strategies not related to the GI. |

**Table 3.** *Cont.*

| Sustainable Development Goals | GI's Potential Contribution | |
|---|---|---|
| | **Straightforward Benefits** | **Ancillary Advantages and Required Actions** |
| | **HOLISTIC IMPACT** | |
| **11 SUSTAINABLE CITIES AND COMMUNITIES** | Turn production constraints (e.g., isolation and weak infrastructures) into assets; Enhanced cross-sector relationships (especially for slow-tourism and gastronomy, beyond a seasonal approach); Promotion and proudness of regional identity, local culture, and natural heritage. | Cross-industry potential not fully unleashed;Limited economic, social, and environmental links between urban and rural areas. |
| **12 RESPONSIBLE CONSUMPTION AND PRODUCTION** | More reliable information to consumers; Fair return for producers; Price premium and better income for the value-adding characteristics of products; De-commoditization (reduced price volatility). | Lack of understanding of detailed meaning of GI schemes;Positive economic impacts highly dependent on economic environment and strategies implemented by operators;Transparency (e.g., via information on website or labels or sustainability reports) hampered for many reasons (data collection, costs, etc.);Scant attention to packaging and labeling methods. |
| **17 PARTNERSHIPS FOR THE GOALS** | Prominent role of the Consortium; Positive "emulation" effect within the Consortium; Partnerships among producers to mobilize and share knowledge, skills, and technology; Partnerships with local actors, universities, and peers located abroad through the Consortium. | Consortium's role to be reinforced;Limited number of effective public–private partnerships, also involving civil society, to achieve the SDGs;Cross-industry partnerships (especially for circular and bioeconomy) little explored;No legislative requirements to guide partnership agreements;Strong individualism and fragmentation;Discrepancies in cultural and entrepreneurial mindsets;Need to increase mutually beneficial partnerships with big players. |

Overall, the results revealed that the "three pillars paradigm" [60] can be an effective basis for clustering the "sustainable" product specification considering that winegrowers follow very practical and straightforward reasoning based on trade-offs between economic, social, environmental, and holistic parameters.

Starting from the economic dimension, the primary data collected confirmed that quality schemes are a valuable tool for fostering sustainable development in rural areas. According to more than half of the respondents, the aggregated promotion of local wine allowed overcoming the strictly local dimension of production, projecting local wine into an international arena that would have otherwise been unattainable. To amplify these results and foster sustainable economic benefits, 64% of the interviewees advocated for the introduction of economic incentives based on environmental commitments together with the reinforcement of physical and digital infrastructures.

Coming to the social pillar, although 57% of the respondents reported the positive impact of GIs, the aggregated data partially contradict this statement, highlighting the need for specific requirements. Indeed, in terms of local employment creation, only 35% of companies hired new people following the adoption of GIs, and only in three cases, the workers were aged 24 or less and/or in vulnerable conditions. Competence is generally preferred over other factors, and although "in the job" training and inter-generational knowledge exchange occur in the majority of the companies (64%), such actions were defined by one interviewee as "necessary burdens". The improvement of the working conditions (declared by 71% of respondents) ultimately depends on the company strategy rather than on the adoption of the GI and does not automatically entail a positive impact on wages (reported by 50% of respondents). The recruitment of workers of different ethnicities and religions was described by all interviewees as a routine practice for the last 15 years, albeit equality and non-discrimination in terms of wages are ensured only by 80% of them. With reference to gender equality, in four companies, the female presence in managerial positions was simply linked to family legacy reasons. Three respondents also declared that gender policies and equality were inherent to the company, so specific policies were not needed.

As far as the environmental dimension is concerned, preserving the natural environment and local resources (landscapes, soils, biodiversity, local genetic varieties) were the most common answers (64–71%). Still, 42% of the interviewees stated that these actions do not relate to the adoption of GIs but rather to the EU Regulation 2018/848 on organic production [61]. The same is true for the circular re-use of marc and semi-solid products (71%), which is carried out for economic reasons as well as to comply with the existing national legislation [62]. Remarkably, two of the interviewees believe they are too small to have any impact on the environment, while another respondent argued that the public rather than the private sector should take responsibility for conservation issues. These elements undoubtedly suggest that the introduction of environmental requirements in the GI product specifications would be of the essence also considering the difficulties related to climate change, which were reported by half of the respondents.

In holistic terms, the role of networks and their implications for sustainable production was acknowledged. The Consortium was defined by two companies as "crucial" for the enhancement of local production from an economic and environmental point of view. Remarkably, almost 30% of the interviewees believe that SDGs implementation shall be delegated to the Consortium only, mainly due to the lack of budget. A pitfall, reported by half of the respondents, jeopardizing the positive "emulation effects" triggered through the Consortium is the strong individualism and fragmentation that characterizes entrepreneurship in the area and makes it difficult to convince people with diverse cultural backgrounds to overcome reticence and enact joint actions.

The community dimension appears to be very much valued: more than half of the interviewees allocated a budget to protect and safeguard local heritage, conceived as a vehicle to promote the cultural message tied to the territory and reinforce the uniqueness of the wine. The restoration of public roads surrounding the production facility was seen

by two companies as a prerequisite to implementing slow oeno-tourism projects. The same respondents also emphasized the environmental benefits deriving from the reconstruction of traditional dry-stone walls, which restored biodiversity.

## 5. Discussion

In a global context marked by deterritorialization, free circulation of people and products, and appraisal of cultural differences, this research aimed to ground the proposition that extending *sui generis* GIs to fashion items would be a favorable solution for both communicating the genius loci and promoting sustainable practices. This process would go hand in hand with the creation of shorter and more responsible supply chains, deeply connecting the wearer and the local fields and pushing for individual or community agency. Opposite to the food industry, though, in the fashion sector, the path could be set in the reverse direction, with GIs being granted to companies not only because of the local scope of production but also considering the abidance by specific sustainability criteria.

### 5.1. Socio-Terroir and Sustainability Criteria for Geographically Rooted Apparel and Footwear

Relying on the product specification found in GIs and collective trademarks operating in the fashion field in France and Italy, respectively, this section outlines additional criteria that could enable "sustainable GIs" in fashion to work. Such principles reflect the growing importance of the socio-terroir elements as well as the social, environmental, and holistic dimensions related to sustainable production and consumption and also based on the cross-industry findings related to sustainability in winemaking.

#### 5.1.1. Raw Materials

Raw materials in fashion are hardly sourced and processed at the local level, making this requirement not crucial for obtaining a GI.

This consideration reflects the prominence of the socio-terroir concept to identify specific non-agricultural products from non-localized variants and aligns with the arguments outlined by Zappalaglio et al. [63]. Indeed, the new quality schemes envisaged by the EU Commission should be based on *sui generis* GIs, namely on Protected Designations of Origin ("PDOs") and Protected Geographical Indications ("PGIs"). It is worth noting that Article 17 of the Regulation also defines Traditional Specialties Guaranteed. However, the EU Commission recognized that, in 28 years, the TSG has not delivered the expected benefits for producers and consumers, and thus, it will be replaced by a more effective and flexible labeling mechanism managed by member states. Therefore, this scheme will not be examined in this paper. The cited study by Zappalaglio et al. suggests that while PDOs would apply only to a limited number of non-agricultural products, PGIs may be more suitable to encompass specifications based both on reputation and on the traditional character of the production method. Indeed, for PGIs, Article 5(2) of Regulation 1151/2012 [64] requires only one step of the production to be completed in the selected area, and the intangible reputational element is generally associated with the history and socio-economic importance of the product.

This proposition is also shared by the French law, where Article L721-2 draws on the EU definition of PGIs and establishes that the GI designates non-agricultural goods originating from a particular territory and possessing specific characteristics that can be essentially attributed to that geographical origin. Article L721-7 No. 4 stresses the "savoir-faire" criteria, acknowledging that a product can be bound to a geographical area by virtue of human rather than environmental factors, such as the traditional know-how or production methods.

This argument seems to be confirmed from a comparative perspective, too, notably taking the Indian GI system [65] into account, where the definition of GIs recalls the one in place in the EU for PGIs. Indeed, pursuant to Art. 2 (e) Geographical Indications of Goods Act, "Geographical indication", in relation to goods, means an indication that identifies such goods as agricultural goods, natural goods, or manufactured goods as originating or

manufactured in the territory of a country or a region or locality in that territory, where a given quality, reputation, or other characteristics of such goods is essentially attributable to its geographical origin [ . . . ]. Considering that India is well-known for protecting more handcrafts than agricultural products [66], it can be inferred that a PGI-like system could be suitable for fashion items as well.

Although PGIs can be more suitable in general terms, this study argues that there is also untapped potential for PDOs. Prominent examples, such as the Fibershed project [67], could be integrated with the creation of bio-based fibers derived from local waste (food and agriculture in particular). Solutions to scale conversion of local biomass into raw materials as well as cross-industry collaborations among local actors operating in different sectors are currently underdeveloped but could optimize the use of neglected resources and revert the fashion industry's extractive pattern by generating raw materials at the local level.

### 5.1.2. Production Methods

Besides the description of the manufacturing practices, product specifications could highlight connections between environmental and human factors as well as holistic benefits related to the use of traditional techniques, the small production scale, and the high customization. Investments and incentives for fostering research and innovation could help overcome the lack of infrastructures for converting proximity fiber production and strategic grazing into fiber and fabric.

### 5.1.3. Link with the Geographic Area

This requirement could be pushed forward by spurring the implementation of closed-loop systems together with rules for locally produced items to be locally processed at the end of their lifecycle, as in the Cardato example.

### 5.1.4. Applicants

Despite the high geographical proximity, the fashion environment is characterized by fragmentation, as each company seems to operate independently from peers located in the same territory. Positive aggregation outputs and the added value stemming from the shared identity embodied in the GI also depend on the degree of representativeness of the applicant as well as on its capacity and commitment to be a driving force for activating sectoral changes towards sustainability and innovation. Therefore, the new Consortium should build trust and credibility while ensuring constant involvement and coordination to overcome cultural barriers, which emerged as one of the main obstacles to translating sustainable commitments into actions. The first step is to ensure the inclusion of all the actors from the entire supply chain, following the French and Swiss models and going beyond the representation requirement, which is not clearly set out in the existing GI regulation at the EU level [68]. This would, to a certain extent, also remedy the imbalance of power affecting the fashion industry [69], often amounting to the marginalization of small entities, and lead to a "coo-petitive" approach [70].

### 5.2. Sustainable GIs and the Benefits of a Trust Enabler in the Consumer, Producer, and Institutional Dimensions

Starting from the socio-terroir elements, *sui generis* GIs would better clarify the influence of environmental and human factors on apparel and footwear coming from a particular area. Highlighting the provenance of a product, this IP asset could operate as a trust enabler in several domains.

Firstly, GIs are recognizable and harmonized signs that consumers are already familiar with as extensively seen on agri-food products. Sustainability credentials could be easily conveyed by differentiating GIs' logo colors based on the level attained, as put forward by the EU Commission in the recent pilot study on product information [71], and/or be coupled with track and trace technologies and advanced product labeling [72]. Such solutions would certainly represent a gain for a sector that has lacked reliable guidelines to

inform sustainable production so far and which primarily relies on multiple and competing certification marks that are largely obscure for consumers [73], incompletely disclosed, and subject to change at the certifiers' whim [74].

The formalization of the genius loci could, in turn, act as a driving force for sustainable tourism development, propelling the involvement of local communities and opening new bottom-up opportunities based on collective fashion heritage, in line with the UNWTO Framework Convention on Tourism Ethics (Article 7, paragraph 2) [75].

GIs would not only increase consumers' trust but also trust within producers through a system that has the potential to involve all actors in the value chain and reward their collective efforts in building up the know-how from generation to generation [76]. To embark producers on the sustainable GI journey, an opening wedge is represented by the economic leverage, provided that the observed or expected impacts of GI systems from producers' perspectives are mainly linked to economic issues rather than environmental benefits—as put forward back in 2009 by Barjolle et al. [77]. Another potential entry point draws on the lack of legal instruments to protect traditional know-how and local craftmanship. GIs could play a significant role by rewarding the community knowledge and, to a certain extent, contribute to curbing cultural appropriation [78]. This, in turn, may lead to greater attention to local resources, namely artisanal skills, as an emblem of cultural identity, which are going to be lost if not adequately passed down to youngsters, with the ancillary benefits of local employment creation and social cohesion.

On another note, horizontal and reciprocal check and balance mechanisms to monitor the quality and sustainability performance would spontaneously take place within the producers' group. Since GIs perform a signaling function regarding the characteristics of the origin-labeled product, all the producers within the group have an incentive to maintain the quality of the product and the sustainability performance as highly as possible.

As a last remark, the sui generis GI system, including sustainability criteria, would increase trust in institutions, starting from the Consortium. The collective organization could also foster interrelations among the GI producers' group to vehiculate strategic initiatives. Sustainability champions, acting as collectors of best practices with a tangible and easily replicable slant, could be identified to demonstrate to peers that a new profitable business model is possible, thus triggering positive emulation effects without leading to the supremacy of most influential producers.

It is also worth noting that GIs would not compromise the heritage and distinctiveness of a brand, which is currently seen as a barrier to collective actions. Indeed, the sign would operate as the lowest common denominator, reflecting an aggregated structure that ensures a stronger image and visibility, deriving from a joint effort. In parallel, each company could keep relying on the goodwill and reputation linked to its own individual trademarks, guarding the related market share.

## 6. Conclusions and Research Limitations

This paper argues that *sui generis* GIs can be the pivot to designate geographically rooted fashion productions and, at the same time, to showcase the sustainable credentials of such products. However, this result would not be obtained automatically; policymakers should address different issues, outlined in the previous sections, regarding how to organize production, adequately motivate producers, create effective connections, and significantly transpose them into product specifications. All these issues sit within the broader "localization" framework and require the implementation of cross-cutting strategies for consolidating the relationship between territories and local stakeholders.

This research, built on a unique cross-sector and transdisciplinary perspective, allows to bring together different but interconnected fields, which do not normally operate in synergy. As such, it grounds the development of policy recommendations that can be useful for grounding the expansion of *sui generis* GIs, provided that the local wine industry is any predictor [79].

The findings may also contribute to the literature engaged in the future evolution of the GI regulation in the EU [35] despite the fact that the definition of the most appropriate *sui generis* GI scheme to protect non-agricultural goods is beyond the scope of this article.

Although the exploratory analysis contributes to an under-explored cross-sector and transdisciplinary research field and provides relevant preliminary findings, its limitations should also be borne in mind. As a small-scale study, the restricted sample size and the very local connotation may make it difficult to generalize study findings.

Future research should consider adopting different levels of analysis, especially from a cultural and phenomenological perspective. Further, the relationship between sustainable food and fashion should be further explored to overcome the struggles experienced by consumers in translating the benefits of slow or organic food into garments and footwear [80].

Finally, it will be worth investigating whether GIs in fashion shall be restricted merely to artisanal production or have the potential to expand and cover garments and footwear in general.

**Author Contributions:** Conceptualization, S.C. and S.G.; methodology, S.C. and S.G.; software, S.C.; validation, S.C. and S.G.; formal analysis, S.C.; investigation, S.C.; resources, S.C. and S.G.; data curation, S.C.; writing—original draft preparation, S.C.; writing—review and editing, S.G.; visualization, S.C. and S.G.; supervision, S.G.; project administration, S.C. and S.G.; funding acquisition, S.G. All authors have read and agreed to the published version of the manuscript.

**Funding:** This research received no external funding.

**Informed Consent Statement:** Informed consent was obtained from all subjects involved in the study.

**Data Availability Statement:** The data presented in this study are available on request from the corresponding author. The data are not publicly available in this article to accommodate the word limits.

**Conflicts of Interest:** The authors declare no conflict of interest.

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
