# Peer review of "Sui Generis Geographical Indications Fostering Localized Sustainable Fashion: A Cross-Industry Assessment"

_sustainability, doi:10.3390/su14095251_

Round 1

Reviewer 1 Report

Thank you for the opportunity to review this interesting article. I took interest and pleasure to read this paper. this paper addresses policy implications for extending GIs to the fashion industry. Followings are my comments:

Significance:

  • The scientific content of this paper is correct for me and deserves to be published.
  • The work innovation should be emphasized in the introduction section. I would also kindly ask to cite the very relevant research paper on exact method and heuristics approach to make a comparison to non-linear model, to increase the quality of the work

Please clarify the hypotheses test used in this paper

Scientific soundness:

  • The subject addressed in this paper is relevant.
  • The study has been correctly designed and is technically sound.

Overall evaluation:

  • The English language quality of this paper is globally appropriate and acceptable. However, some minor revisions and spell check seem to be necessary.

As a conclusion, my suggestion to the editor is to accept this paper for publication after minor revision.

Author Response

Please, see attached PDF.

Reviewer 2 Report

The paper on “Sui generis Geographical Indications fostering localized sustainable fashion: a cross-industry assessment” addresses policy implications for extending geographical indications (GIs) to the fashion industry. By taking the Apulia region as a case study, the research assesses to what extent apparel and footwear items are compatible with the scope of protection of the EU sui generis GIs system, despite their non-terroir character. After review, I have comments and suggestions as follows.

1). The research methods and results are not clearly presented in the Abstract section. I suggest rewriting the abstract by clarifying the research methods and results.

2). I suggest adding/indicating the significance of your research in the Introduction section in order to introduce the contributions of your research to readers.

3). As sustainability aspects are among the objectives of your research; therefore, I suggest expanding your literature/theoretical background on related sustainability issues.

4). More elaborating on the case study selection would be best: Why Linge Basque and Charentaise de Charente-Périgord (France) and Pelle Conciata al Vegetale in Toscana and Cardato Recyled Made in Prato (Italy).

5). I did not see any explanation regarding the respondents you interviewed. How many interviewees? How did you select them? Interview procedure?

6). For your results of Tables 1 & 2, even though you put them in the Appendix, I suggest adding citations in the text in that table OR adding another column in that table for putting sources (where are they sourced/adapted from).

7). The conclusion looks too general—seems not a real conclusion. The conclusion in a scientific article should describe the usefulness of the results in the field of research and of course, be limited to the specific area of research investigated.

8). I suggest following the Instructions for Authors for writing your references and for using appendixes. Also, those official documents, legislative sources and case law, and websites could be cited as reference no.

Author Response

Please, see attached PDF.

Round 2

Reviewer 2 Report

The authors took my comments into account and improved their paper.